# Radio Resource Allocation with The Fairness Metric for Low Density Signature OFDM in Underlay Cognitive Radio Networks [note 1]

**DOI:** 10.3390/s19081921

**Published:** 2019-04-23

**Authors:** Linda Meylani, Adit Kurniawan, M. Sigit Arifianto

**Affiliations:** 1School of Electrical Engineering and Informatics, Bandung Institute of Technology, Jl. Ganesha No. 10, Bandung 40132, Indonesia; adit@stei.itb.ac.id (A.K.); msarif2a@stei.itb.ac.id (M.S.A.); 2School of Electrical Engineering, Telkom University, Jl. Telekomunikasi No. 1, Bandung 40257, Indonesia

**Keywords:** resource allocation, low density signature OFDM, underlay cognitive radio networks

## Abstract

Low density signature orthogonal frequency division multiplexing (LDS-OFDM), one type of non-orthogonal multiple access (NOMA), is a special case of multi-carrier code division multiple access (MC-CDMA). In LDS-OFDM, each user is allowed to spread its symbols in a small set of subcarriers, and there is only a small group of users that are permitted to share the same subcarrier. In this paper, we study the resource allocation for LDS-OFDM as the multiple access model in cognitive radio networks. In our scheme, SUs are allocated to certain $d_{v}$ subcarriers based on minimum interference or higher SINR in each subcarrier. To overcome the problem where SUs were allocated less than the $d_{v}$ subcarriers, we propose interference limit-based resource allocation with the fairness metric (ILRA-FM). Simulation results show that, compared to the ILRA algorithm, the ILRA-FM algorithm has a lower outage probability and higher fairness metric value and also a higher throughput fairness index.

## 1. Introduction

To improve the spectrum efficiency, cognitive radio (CR) systems allow the secondary user (SU) to access the primary user’s (PU) spectrum in three schemes of dynamic spectrum access (DSA) [1]; these are interweave, underlay, and overlay. The interweave scheme permits the SU to access the PU’s spectrum if and only if the PU’s spectrum is in idle condition, that is when the PU does not occupy the frequency spectrum. In the underlay scheme, the SU is allowed to access the spectrum simultaneously with the PU, as long as it does not cause any degradation in PU performance. Therefore, the SU must pay attention to the transmission power usage so that the arising interference is still below the interference threshold as determined by the PU. In the overlay scheme, the SU is permitted to transmit simultaneously with the PU provided that the SU allocates part of its transmit power to relay the PU’s signal.

There are many studies related to spectrum access and interference mitigation in CR systems. Due to the flexibility in allocating resource among SUs, many researchers have used multi-carrier transmission in CR networks. The work in in [2,3,4,5] used multi-carrier code division multiple access (MC-CDMA) as their spectrum access model and designed interference cancellation (IC) techniques to combat the interference. Reference [4] assumed that each PU and SU interfered with each other. To combat the interference both in PUs and SUs, the work in [4] used dual list interference cancellation (DLIC), as a combination of successive interference cancellation (SIC) and parallel interference cancellation (PIC). Unlike the previous work in [4], in this paper, we use low density signature orthogonal frequency division multiple access (LDS-OFDM) as a multiple access system and manage the interference felt by each SU to increase the spectrum utilization. As introduced in [6], LDS-OFDM combines two systems: LDS in code division multiple access (CDMA) systems as introduced in [7] and OFDM systems. Due to the use of low density signatures in LDS-OFDM, each user will spread its symbol in a small group of subcarriers, and as a result, each user only receives the interference from a small subset of users.

Research on resource allocation in the LDS system had been carried out in [8,9]. In [8], the water filling (WF) algorithm and equal power (EP) algorithm ware used to allocate subcarrier and power for each user in LDS-OFDM. The simulation results showed that the spectral efficiency of LDS-OFDM, when using the same algorithm, was higher than OFDMA systems. In [9], the authors proposed two algorithms of subcarrier and power allocation for each user in multicarrier low density signature multiple access (MC-LDSMA) based on the WF algorithm and maximum rate transmission.

A comprehensive survey of resource allocation in cognitive radio systems can be found in [10,11]. Moreover, the study of resource allocation in multicarrier-based cognitive radio networks has been given much attention, for both single-user and multi-user cases. In [12], the authors formulated joint power and rate allocation for CDMA users in an underlay scheme by using proportional and max-min fairness as the optimization problems. The resource allocation algorithm for OFDM in underlay CR with decoding strategies at the secondary receiver was conducted by [13]. The author in [14] proposed stochastic resource allocation to optimize the sum-rate performance of a CR network. A CR network in reference [14] was developed by combination of two schemes of DSA, i.e, interweave and underlay. Reference [15] proposed the resource allocation algorithm for LDS-OFDM users based on the interference limit as defined by each PU, while [16] proposed robust resource allocation for the SU based on a graph model. Channel selection in the cognitive radio ad hoc sensor network (CRASN) has been proposed by [17]. Reference [17] proposed an opportunistic channel selection scheme (OCSS). This scheme made CRASN intelligently cater to channel switching and inform the next-hop channel for seamless communication.

In this paper, we propose resource allocation for LDS-OFDM users in underlay CR systems based on the interference limit with the fairness metric to increase spectrum utilization. The use of the fairness metric is intended to increase the fairness of allocating $d_{v}$ subcarriers to each user. A difference from the previous work in [15] is related to the exertion of the $d_{c}$ parameter not only as a limiting parameter, but also as a parameter to find an available subcarrier.

The rest of this paper is organized as follows. Section 2 describes the system model and proposed algorithm to solve the fairness problem of allocated $d_{v}$ subcarriers to each user. Section 3 presents an evaluation of the performance of the proposed algorithm in the perfect channel state information CSI condition. Section 4 provides discussion of the results. Finally, Section 5 gives the conclusion of the paper.

## 2. Materials and Methods

### 2.1. System Model

In this paper, we consider an uplink scenario in an underlay CR network with multiple channels. There exists a set of PUs I={1,2,…,I}, whereby the PUs have priorities to access the $N_{PU}$ spectrum channel from $N_{total}$ available channels and a set of SUs J={1,2,…,J} that are keen to transmit simultaneously. In our system, the PUs’ system used OFDMA as the multiple access and without any interference from other PUs, while SUs used LDS-OFDM. Note here that SUs have a lower transmission power than PUs. Our system model is shown in Figure 1.

Since CR system used LDS-OFDM as its multiple access scheme, each of the users in CR will spread their symbols using the low density signature (LDS). The low density signature refers to a sparse Nchips consisting of $d_{v}$ non-zero elements. $d_{v}$ non-zero elements in LDS chips refer to the representation of the subcarrier mapping of each user. Let $S_{n}$ be the set of users at the $n^{\mathrm{th}}$ subcarrier. In LDS-OFDM, $|S_{n}|\leq d_{c}$ must be satisfied. In the underlay CR scheme, SUs are permitted to access the spectrum simultaneously with the PU provided that the PU’s performance does not degrade below the desired threshold. The PU’s signal and the SU’s signal at the same subcarrier will be superimposed. The received signal in the $n^{\mathrm{th}}$ subcarrier can be stated as follows:(1)$$y_{i,n}=h_{i,n}\cdot a_{PU_{i,n}}+\sum_{j\in S_{n}}g_{j,n}\cdot c_{SU_{j,n}}+v_{n}.$$
where the channels of the $i^{\mathrm{th}}$ PU and the $j^{\mathrm{th}}$ SU at the $n^{\mathrm{th}}$ subcarrier are denoted as $h_{i,n}$ and $g_{j,n}$, respectively. The $a_{PU_{i,n}}$ as the data symbol of the $i^{\mathrm{th}}$ PU is transmitted in the $n^{\mathrm{th}}$ subcarrier. $c_{SU_{j,n}}$ as the chips vector belongs to the $j^{\mathrm{th}}$ SU after the spreading process, and $v_{n}$ refer to an additive white Gaussian noise (AWGN).

The connectivity of 6 users and 4 resources in LDS-OFDM can be depicted as a bipartite graph, as shown in Figure 2. Figure 2 shows that each SU utilizes $d_{v}=2$ subcarriers to transmit their symbols, and each subcarrier is accessed by $d_{c}=3$ different SUs. The relationship of PUs, SUs, and subcarriers in a graph is shown in Figure 3.

### 2.2. Problem Formulation

To formulate the subcarrier allocation problem for LDS-OFDM users in an underlay CR, let $x_{j,n}$ be the resource allocation index; therefore, $x_{j,n}=1$ if $n^{\mathrm{th}}$ subcarrier is allocated to the $j^{\mathrm{th}}$ SU, and $x_{j,n}=0$ otherwise. In LDS-OFDM, by using the spreading, the same message (information) is repeated on the set of subcarriers ($N_{j}$). In this paper, we assume the set of subcarriers used by each SU to consist of $d_{v}$ subcarriers. Consequently, the sum-rate of LDS-OFDM users in underlay CR can be formulated as follows:(2)$$\max\sum_{n\in N_{j}}R_{j,n},$$
where,
(3)$$R_{j,n}=\log_{2}\left(1+\frac{|g_{j,n}{|}^{2}\cdot p_{j,n}}{\sigma^{2}\cdot W_{n}+I_{i,n}^{PU}+\sum_{j\in S_{n}}I_{j,n}^{SU}}\right),$$
subject to
(4)$$\sum_{n\in S_{n}}p_{j,n}\leq P_{SU,\max},\forall j\in J,$$
(5)$$p_{j,n}\geq 0,\forall j\in J,n\in N,$$
(6)$$x_{j,n}\in \left\{0,1\right\},\forall j\in J,n\in N,$$
(7)$$\sum_{j\in S_{n}}x_{j,n}\leq d_{c},\forall n\in N,$$
(8)$$\sum_{n\in N_{j}}x_{j,n}\leq d_{v},\forall j\in J,$$
(9)$$I_{total,n}=\sum_{j\in S_{n}}{\left|g_{j,n}\right|}^{2}\cdot p_{j,n}\leq I_{th,n},\forall j\in S_{n},$$
(10)$$I_{th,n}=I_{th},$$
(11)$$SINR_{i,n}=\frac{|h_{i,n}{|}^{2}\cdot p_{i,n}}{\sigma^{2}\cdot W_{n}+\sum_{j\in S_{n}}I_{j,n}^{SU}}$$
where $p_{j,n}$ denotes the transmission power of the $j^{\mathrm{th}}$ SU at the $n^{\mathrm{th}}$ subcarrier. The noise spectral density, $\sigma^{2}$, is assumed to be the same on all spectrum bands, and each subcarrier has the same bandwidth, $W_{n}$. SUs in LDS-OFDM spread their messages on the set of $d_{v}$ subcarriers, as shown in (Equation 8). Unlike the allocation resource in OFDMA [18], the constraint in (Equation 7) cannot be relaxed by stating the value of $x_{j,n}$ as any real value between 0 and 1. If the constraint in (Equation 7) is relaxed, the concept of low density signature is violated and causes each user to interfere with each other, and automatically, the constraints in (Equation 7) and (Equation 8) cannot be fulfilled. The constraints in (Equation 9) and (Equation 10) provide a protection to PUs where the aggregate interference generated by SUs at the $n^{\mathrm{th}}$ subcarrier must be below the interference threshold $I_{th}$. SINR (Equation 11) is used to find the available subcarrier for SUs. The rate of the $i^{\mathrm{th}}$ PU at the $n^{\mathrm{th}}$ subcarrier is defined as:(12)$$R_{i,n}=\log_{2}\left(1+\frac{|h_{i,n}{|}^{2}\cdot p_{i,n}}{\sigma^{2}\cdot W_{n}+\sum_{j\in S_{n}}I_{j,n}^{SU}}\right).$$

### 2.3. Interference Limit-Based Resource Allocation

Reference [15] already allocated LDS-OFDM users based on the interference limit; we name it as the interference limit-based resource allocation algorithm (ILRA). ILRA allocates a number of resources for each SU based on a lower interference value or a higher SINR in each subcarrier. ILRA is shown in Table 1.

In the first step, we must set the values of $d_{v}$, $d_{c}$, the SINR threshold, the total number of subcarriers, *N*, the number of subcarriers accessed by each PU, $N_{PU}=N_{PU}$, the total number of PU users, *I*, the total number of SU users, *J*, and the transmission power both of the PU and SU, $P_{PU}$ and $P_{SU}$. After all PUs have accessed their subcarriers, the CR system would calculate the interference or SINR in each subcarrier and sort the subcarriers based on the highest SINR. Select $d_{v}$ subcarriers with the highest SINR to be allocated to the $j^{\mathrm{th}}$ SU provided that the subcarriers meet the requirement (Equation 7). If there are dx subcarriers in $d_{v}$ selected subcarriers that have been accessed by $d_{c}$ users, the $j^{\mathrm{th}}$ SU will only obtain an allocation of $d_{v}-dx$ subcarriers. Hence, ILRA causes the number of the resource allocations for the $j^{\mathrm{th}}$ SU to be less than $d_{v}$ subcarriers.

### 2.4. Interference Limit-Based Resource Allocation with Fairness Metric Algorithm

In this paper, we assume that fairness is achieved if all SUs are allocated with the same number of resources; therefore, the fairness metric (FM) for SUs can be defined as follows:(13)$$FM=\frac{\sum_{j}SU_{j,d_{v}}}{J}$$
(14)0≤FM≤1
where $SU_{j,d_{v}}$ represents the $j^{\mathrm{th}}$ SU with the allocated the $d_{v}$ subcarrier and *J* as the maximum number of SUs. Therefore, in this paper, we propose the interference limit-based resource allocation with fairness metric (ILRA-FM) algorithm. The value of 1 for the fairness metric indicates that the CR system successfully allocates the $d_{v}$ subcarrier for each SU. In this paper, we also calculate the value of fairness index (FI), describing fairness on throughput allocation for each SU [12]. The fairness index value is defined as:(15)$$FI=\frac{{\left(\sum_{j=1}^{J}R_{j}\right)}^{2}}{J(\sum_{j=1}^{J}R_{j}^{2})},$$
where $R_{j}$ is the throughput of the $j^{\mathrm{th}}$ SU and *J* is the maximum number of SUs. The rate allocation becomes fairer if the value of FI is close to one. Our proposed ILRA-FM algorithm is described in Table 2.

The first step in the ILRA-FM algorithm is setting the value of all parameters. The second step is the initiation process. In this step, all PUs access their channels and the CR system will calculate the SINR for each subcarrier based on (Equation 11) by assuming there is no SU in each subcarrier ($I_{i,n}=0$). By using two parameters (*d* and d1) in the ILRA-FM algorithm, CR determines the number of $d_{v}$ subcarriers that have higher SINR and checks for the constraint in (Equation 7). If the constraint in (Equation 7) has been fulfilled, the system will allocate $d_{v}$ subcarriers to the SU, calculate the interference in each subcarrier based on (Equation 9) and update the SINR value for each subcarrier based on (Equation 11). However, if one or several selected subcarriers (based on parameter *d*) do not meet the constraint (Equation 7), the system will search for available subcarriers that meet Constraint (Equation 7) based on d1, check the constraint (Equation 9) for each selected subcarrier, allocate the subcarriers to the SU, and update the SINR value. Figure 4 illustrates a timing diagram for the operational example of the ILRA-FM algorithm when $d_{v}=2$ and $d_{c}=3$. The value “1” in the allocated subcarriers indicates that the subcarriers are allocated to the SU. At $t=t_{0}$, the system calculates the SINR in each subcarrier. Based on the highest SINR value, the *d* parameter will choose $d_{v}$ subcarriers; in Figure 4, the third and the fourth subcarriers are selected, and the first and the second subcarriers are selected based on parameter d1. Although there are 4 different subcarriers to be selected based on *d* and d1, the ILRA-FM algorithm always chooses $d_{v}$ subcarriers based on *d* for the first step, except the selected subcarriers that are not adequatefor (Equation 7). Therefore, at $t=t_{1}$, the third and fourth subcarriers are allocated to SU1, while based on the selected subcarrier at $t=t_{5}$, the first and the second subcarriers are allocated to SU6. Although the third subcarrier is selected (based on *d*), it is not appropriate for Constraint (Equation 7), because it has been accessed by three different users, SU1, SU2, and SU4. Due to the value of the $d_{c}$ parameter being equal to 3, consequently, SU8 is allowed to access one subcarrier only (second subcarrier), and SU9 is not allowed to access any subcarrier.

The two types of ILRA-FM algorithm have the same goal, i.e., to minimize the outage probability and increase the fairness index. Similar to ILRA-FM Type 1, ILRA-FM Type 2 also use two parameters, *d* and d1. In ILRA-FM Type 2, the selected subcarrier from d1 is always compared to the selected subcarrier from *d* in terms of the number of users accessing it. Therefore, the allocated subcarriers are not always prioritized based on the highest SINR.

## 3. Results

To evaluate the performance of the proposed algorithm, we considered the case with 10 PUs and 200 SUs. The maximum SU’s transmit power and the maximum PU’s transmit power were 10 dBm and 23 dBm, respectively. The bandwidth in each subcarrier was 15 kHz, and there were 100 subcarriers in the whole system. The link attenuation between the base-station and a user (PU or SU) was provided by the Cost231-wiNLOS path loss model. The noise spectral density $\sigma^{2}$ was assumed to be −300 dBm/Hz. The simulation parameters are shown in Table 3.

As for the performance evaluation metric, we utilized the outage probability (OP), the average throughput for the SU and PU, the fairness matric (FM) value, and the throughput fairness index (FI). The outage probability is a parameter stating the number of SUs that do not obtain a resource allocation divided by the total number of SUs. The OP values are in the range of 0–1. The value of OP will be equal to zero (OP=0) when all SUs obtain a resource allocation, and the OP will equal one (OP=1) when all SUs have not obtained any resource. The value of the average throughput for PU and SU is affected by a number of allocated SUs in each subcarrier. Based on (Equation 3) and (Equation 12), the increasing number of allocated SUs in each subcarrier will decrease the value of the average throughput. If FM determines fairness on the number of allocated resources for each SU, FI states fairness on throughput allocation for each SU.

### 3.1. Performance of the Proposed Algorithm

Figure 5 shows the outage probability, the number of SUs that obtained the resource allocation of less than $d_{v}$ subcarriers, the fairness metric, and the throughput fairness index for the ILRA algorithm and the ILRA-FM algorithm Types 1 and 2 with all algorithms setting the same value of $d_{v}$ and $d_{c}$. Figure 5a shows that the proposed algorithm (ILRA-FM-1 and ILRA-FM-2) had lower outage probability than the ILRA algorithm as proposed in [15]. The addition of the sorting parameter d1 in the ILRA-FM algorithm caused the algorithm to ensure that each SU acquired $d_{v}$ subcarriers by searching for alternative subcarriers accessible for the SU when the selected subcarriers based on SINR did not fulfill the constraint in (Equation 7). Consequently, a number of SUs obtaining the resource allocation will increase and cause a decreasing outage probability (as shown in Figure 5a), while increasing the fairness metric (Figure 5c) and the throughput fairness index (shown in Figure 5d). Figure 5b states the comparison of the number of users with resource allocation less than $d_{v}$ subcarriers to the total number of users in the system. This figure shows that ILRA FM Types 1 and 2 had better performance than ILRA in terms of allocating $d_{v}$ subcarriers on each SU. However, ILRA-FM Type 2 had a lower performance than Type 1 in the number of subcarriers allocated to each SU. ILRA-FM Type 2 always compared the selected subcarriers based on the parameters *d* and d1 in terms of the number of users who have accessed the subcarrier. When the selected subcarriers of the two parameters differed in sequence, for example the selected subcarrier based on *d* was SC1 (accessed by two users) and SC3 (accessed by four users), and the selected subcarrier based on d1 SC2 (accessed by three users) and SC1 (accessed by two user) will be allocated to the $j^{\mathrm{th}}$ SU. In the case of $d_{v}=2$, ILRA-FM Type 2 will compare SC1 to SC2, choose SC1 because SC1 was only accessed by one user, and update the number of users accessing SC1 to three users; when ILRA-FM Type 2 compares the next selected subcarrier, SC3 and SC1, the algorithm still chooses SC1 and causes the number of subcarriers allocated to the $j^{\mathrm{th}}$ SU to be only SC1. However, ILRA-FM Type 2 still had a better value of the fairness metric than ILRA, as shown in Figure 5c.

Figure 6a,b shows the average throughput both of PUs and SUs. From the figures, it can be seen that the ILRA-FM algorithm had a lower throughput value than the ILRA algorithm. The low throughput value in the ILRA-FM algorithm was caused by the higher interference value perceived by each user, both PUs and SUs, as a result of the increase in the number of SUs allocated in each resource. The average throughput shown in Figure 6 only takes into account the average throughput of allocated users, while the throughput fairness index in Figure 5d considers the throughput of all SUs. Therefore, the proposed ILRA-FM algorithm Types 1 and 2 in Figure 5d have a higher fairness index than the ILRA algorithm.

### 3.2. Influence of $d_{c}$ and $d_{v}$

Figure 7, Figure 8 and Figure 9 show the effect of different values of $d_{c}$ when the system set the fixed value of dv. Setting a fixed value on $d_{c}$ in the LDS-OFDM system causes the number of SUs who can access each resource to be limited to $d_{c}$ users; therefore, the smaller value of $d_{c}$ will increase the outage probability, and higher $d_{c}$ will cause more SU users to be allowed to access the subcarrier. Therefore, the outage probability will decrease, and the fairness metric and throughput fairness index will increase. Figure 7 proves that a higher value of $d_{c}$ has a minimum OP compared to another value of $d_{c}$ (four and 12). Figure 8a,b shows that a higher value of $d_{c}$ causes higher FM and FI, as well. The smaller value of $d_{c}$ also caused both the PU’s and SU’s transmission rate to be higher due to low interference felt by each user. Figure 9a,b shows the average throughput of both PUs and SUs. From these figures, it can be stated that a higher value of $d_{c}$ will cause lower throughput for both PUs and SUs. As a higher number of SUs access each subcarrier, the interference felt by each user, both PUs and SUs, will be higher. Based on (Equation 4) and (Equation 14), the higher interference will reduce the throughput in each subcarrier.

Figure 10 shows the effect of the $d_{v}$ parameter on the outage probability. Figure 11 shows the effect of the $d_{v}$ parameter on the fairness metric and throughput fairness index; while Figure 12 shows the effect of the $d_{v}$ parameter on the average throughput of the PU and SU. The higher value of $d_{v}$ caused a higher outage probability, a lower fairness value on FM and FI, and also a lower average throughput for both PUs and SUs. The $d_{v}$ parameter states the number of subcarriers accessed by each SU and determines the SU’s transmission power in each subcarrier. When the $d_{v}$ parameter is set high, then each SU will spread its symbols to more subcarriers and cause the number of users accessing the subcarrier to reach the limit faster; then, the outage probability will increase, and both fairness values will decrease. Due to the higher value of $d_{v}$, the SU’s transmission power on each subcarrier will be lower. Although the higher value of $d_{v}$ causes the number of users accessing the subcarrier to be much closer to $d_{c}$, due to the lower transmission power in each subcarrier, the SU’s transmission rate will be lower. The effect of $d_{v}$ parameters on average throughput is shown in Figure 12. Figure 12a shows the average throughput of the PU. When the number of active SUs on the system is in the range of 20–100 users, the higher $d_{v}$ value caused lower throughput of the PU. Conversely, when the number of users was greater than 100 users, the lower $d_{v}$ value caused lower throughput of the PU. The decreasing average throughput of the PU was caused by the increased interference felt by the PU. When $d_{v}$ is set small, the transmission power of the SU in each allocated subcarrier will be greater. Figure 11a shows that the fairness metric values were almost close to one when the number of users was 100 and $d_{v}=4$, meaning that each SU will attain the same number of subcarriers. Because of the system set $d_{c}=9$, each subcarrier was accessed by a dcuser (nine users), which means that there were nine SUs with higher transmission power accessing the same subcarrier.

## 4. Discussion

In our paper, we assumed that each SU had the same priority. Therefore, we proposed the interference limit-based resource allocation with fairness metric (ILRA-FM) algorithm to find an available subcarrier accessible for each SU and to ensure each SU in the CR system had the same number of subcarriers ($d_{v}$). When SUs spread their message in the same number of subcarriers ($d_{v}$) and each subcarrier is accessed by the same number of SUs ($d_{c}$), LDS-OFDM will have a regular mapping matrix. The decoding process in the receiver will use the message passing algorithm (MPA) [7]. The complexity of MPA decoding is influenced by the value of the dv and $d_{c}$ parameters. A greater the value of $d_{c}$ and $d_{v}$ causes a higher complexity in the decoding process.

In our proposed ILRA-FM algorithm, there were four important things that must be fulfilled to ensure that the secondary users were able to access a subcarrier, i.e., (1) the desired interference threshold value (defined by PU) in each subcarrier ($I_{th,n}$) and (2) the number of subcarriers that can be accessed by each user to transmit their symbol ($d_{v}$), (3) the number of users permitted to access the same resource ($d_{c}$) and (4) the SU’s transmission power.

Our simulation results in Figure 5 and Figure 6 compared the performance of our proposed ILRA-FM algorithm with the previous work ILRA in [15]. In these figures, it can be seen that the ILRA-FM algorithm had lower outage probability and a higher fairness metric and higher throughput fairness index compared to the previous works. The lower transmission rate of the PU and SU in the ILRA-FM algorithm was caused by the larger number of SUs allocated in each resource, causing the ratio of the transmission rate to interfering rate to be smaller.

The highest value of the outage probability was obtained when the CR system used the smallest $d_{c}$ value. This occurred because the system limited the number of users allowed to access the existing resource ($d_{c}$), even though the interference that appeared in the resource was still under the interference threshold. When the system configures a low value of $d_{c}$, only a small number of SUs will obtain resources, and it will cause a high outage probability. Although ILRA-FM Types 1 and 2 had a higher value of fairness index than ILRA, fairness on throughput allocation for SUs was still low (FI≪1). Figure 8b and Figure 11b show that the value of FI was influenced by $d_{v}$ and $d_{c}$. A higher value of $d_{c}$ caused a large number of SUs to obtain resource allocation. Although Figure 8a shows that the value of FM was closer to one (20≤J≤60), however the FI value in Figure 8b was still lower than 0.4. This means that channel gain perceived by each SU had a different value; therefore, it will affect the acceptability and rate of each SU.

The higher value of $d_{v}$ indicates a greater number of resources used by the SU to spread its symbols and makes more resources update the value of $I_{k,n}$. The more accessed resource can reach the $d_{c}$ boundary faster and cause a higher OP value. When the CR system sets a small value of $d_{v}$, the SU’s transmission power in each subcarrier will be higher than the SU’s transmission power when the CR system sets a higher value of $d_{v}$. As a result of the large transmission power on each subcarrier, the perceived interference felt by the PU on each subcarrier accessed by the SU will be greater and cause a lower transmission rate of PUs.

## 5. Conclusions

The proposed interference limit resource allocation with fairness metric (ILRA-FM) algorithm successfully allocated multiple resources to multiple SUs in the LDS-OFDM underlay CR. Simulation results showed that compared to ILRA, the ILRA-FM algorithm had a lower outage probability than ILRA, a higher value of the fairness metric, and also a higher throughput fairness index. Simulation results also showed that both PUs and SUs in ILRA-FM had lower average throughput compared to the ILRA algorithm due to the increasing interference in each subcarrier as a result of increasing the number of allocated SUs. Using two parameters based on the interference limit or the PU’s SINR in each subcarrier ($SINR_{PU,n}$) and the number of SUs that accessing each subcarrier can increase both the fairness value, the fairness metric, and the throughput fairness index. Setting a higher value for the $d_{c}$ parameters can decrease the outage probability; on the other hand, this will cause higher computational complexity in the decoding process. Setting a lower value in $d_{v}$ will make the outage probability decrease and cause the fairness metric and throughput fairness index to increase.

## Figures and Tables

**Figure 1 sensors-19-01921-f001:**
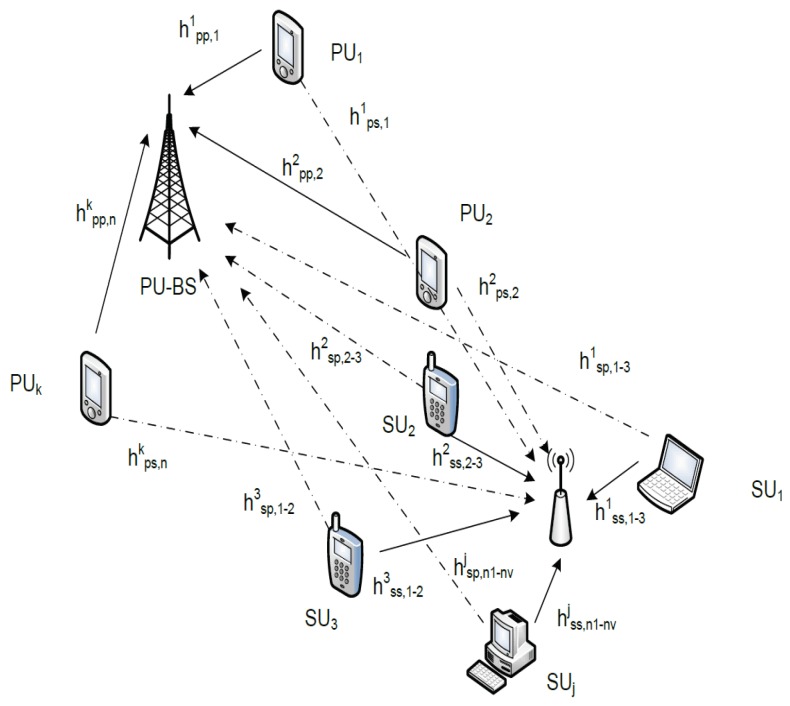
System model.

**Figure 2 sensors-19-01921-f002:**
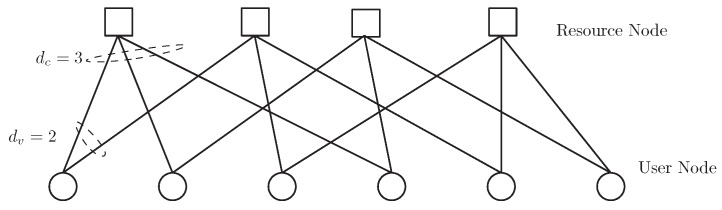
Bipartite graph of 4 resources and 6 users.

**Figure 3 sensors-19-01921-f003:**
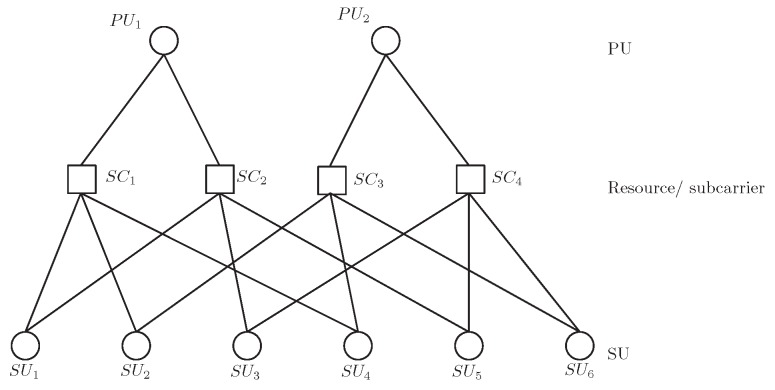
The relationship of PUs, SUs, and resources in a graph.

**Figure 4 sensors-19-01921-f004:**
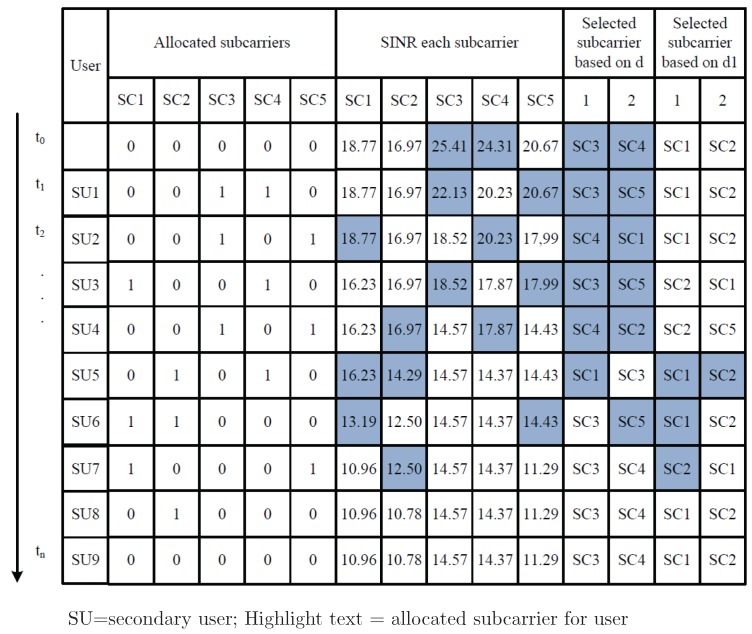
Timing diagram for the operational example of the ILRA-FM algorithm Type 1.

**Figure 5 sensors-19-01921-f005:**
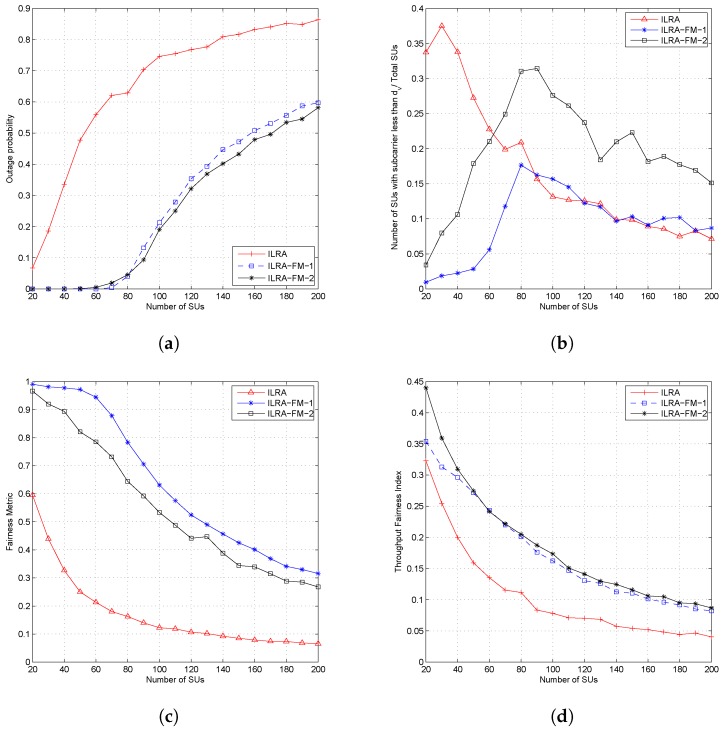
Performance of the proposed algorithm at $d_{c}=6$ and $d_{v}=8$, (**a**) outage probability, (**b**) number of SUs that obtain less than $d_{v}$ subcarriers, (**c**) fairness metric, and (**d**) throughput fairness Index.

**Figure 6 sensors-19-01921-f006:**
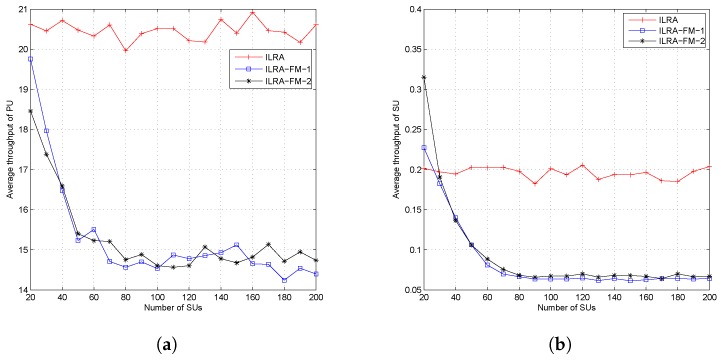
Throughput of the proposed algorithm at $d_{c}=6$ and $d_{v}=8$; (**a**) PU and (**b**) SU.

**Figure 7 sensors-19-01921-f007:**
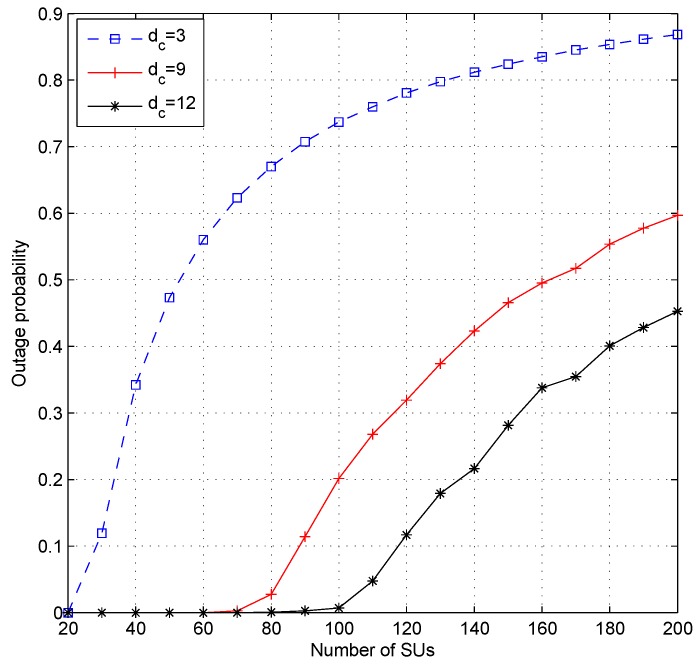
Outage probability of the proposed ILRA-FM Type 1 algorithm with variation of the value of $d_{c}$ and fixed $d_{v}$ ($d_{v}=12$).

**Figure 8 sensors-19-01921-f008:**
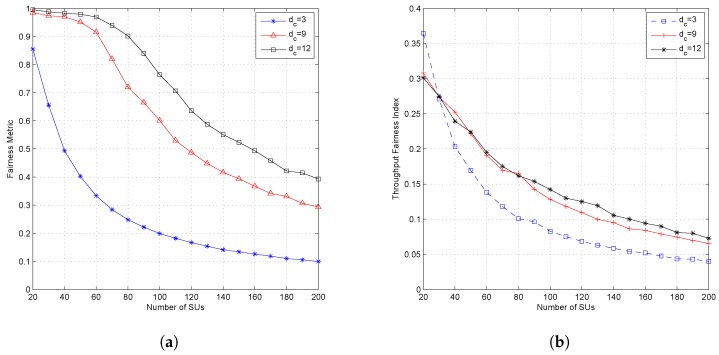
Performance of the proposed algorithm with variation of the value of $d_{c}$ and fixed $d_{v}$ ($d_{v}=12$); (**a**) fairness metric and (**b**) throughput fairness index.

**Figure 9 sensors-19-01921-f009:**
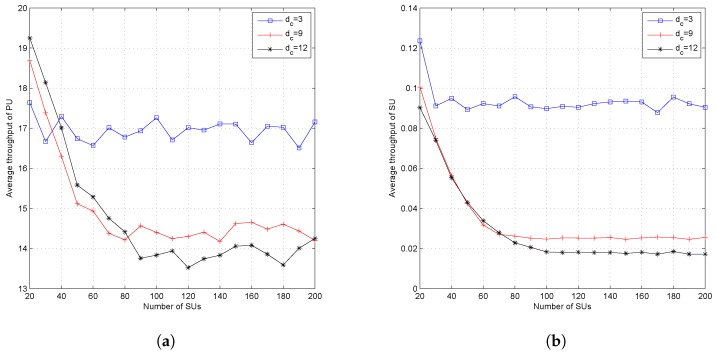
Performance of the proposed algorithm with variation of the value of $d_{c}$ and fixed $d_{v}$ ($d_{v}=12$); (**a**) throughput of PUs and (**b**) throughput of SUs.

**Figure 10 sensors-19-01921-f010:**
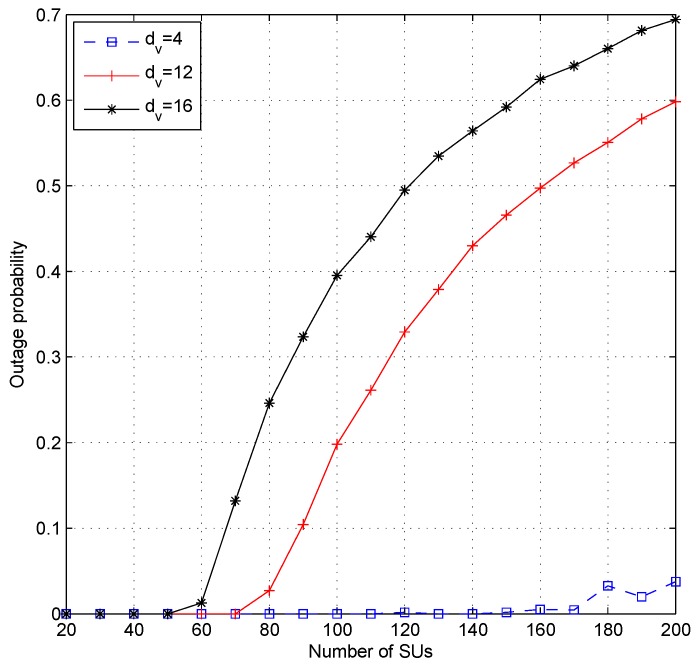
Outage probability of the proposed ILRA-FM Type 1 algorithm with the variation value of $d_{v}$ and fixed $d_{c}$ ($d_{c}=9$).

**Figure 11 sensors-19-01921-f011:**
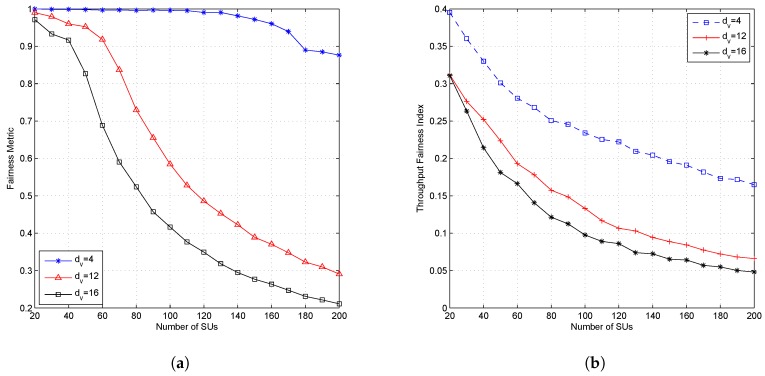
Performance of the proposed algorithm with the variation value of $d_{v}$ and fixed $d_{c}$ ($d_{c}=9$); (**a**) fairness metric and (**b**) throughput fairness index.

**Figure 12 sensors-19-01921-f012:**
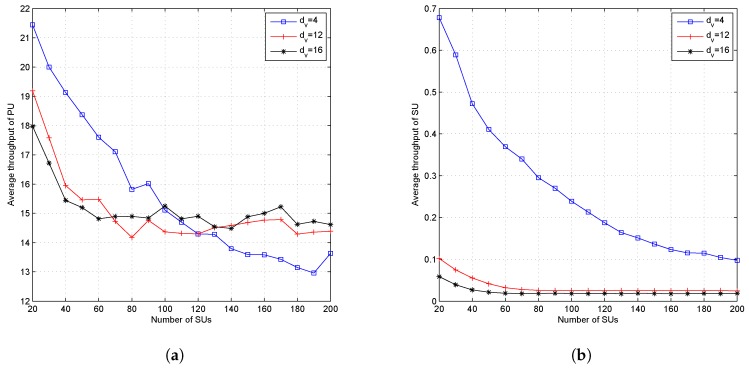
Performance of the proposed algorithm with variation of the value of $d_{v}$ and fixed $d_{c}$ ($d_{c}=9$); (**a**) throughput of SUs and (**b**) throughput fairness index.

**Table 1 sensors-19-01921-t001:** ILRA.

1:	**Input the variable:**$d_{v}$, $d_{c}$, and $\Gamma_{th}=\Gamma_{PU}$, *N*, $N_{PU}=N_{PU}$, *I*, *J*,
	$P_{i,n}=\frac{P_{PU}}{N_{PU}}$, $P_{j}=P_{SU}$, $I_{i,n}=0,\forall n$.
2:	**Initiation**: Each PU accesses its own subcarrier
	Counting SINR in each subcarrier ($\Gamma_{n}$) based on (Equation 11)
3:	$\forall SU_{j}$, find the $d_{v}$ subcarrier that has higher SINR ($\Gamma_{n}$) or lower interference
4:	$\forall d_{v,n}\in d$ according to the requirements
	**If**$\Gamma_{v,n}>\Gamma_{th}$, **and** (Equation 7) id fulfilled
	Allocate the $n^{\mathrm{th}}$ subcarrier to $SU_{j}$
5:	Calculate: Interference on each of the $n^{\mathrm{th}}$ subcarriers based on (Equation 9)
6:	Update ($\Gamma_{n}$) (Equation 11)
7:	**End if** $\sum_{SU,n}=d_{c}$ **or** $\Gamma_{n}\leq \Gamma_{th}$
	**or**∀SU have been allocated.

**Table 2 sensors-19-01921-t002:** ILRA-FM algorithm (Type 1).

	Proposed algorithm to minimize **OP** and increase **FI**
1:	**Input the variable**: $d_{v}$, $d_{c}$, and $\Gamma_{th}=\Gamma_{PU}$, *N*, $N_{PU}=N_{PU}$, *I*, *J*,
	$P_{i,n}=P_{PU}$, $P_{j}=P_{SU}$, $I_{i,n}=0,\forall n$.
2:	**Initiation**: PU accesses its own subcarrier
	**Counting** SINR in each subcarrier ($\Gamma_{n}$) based on (Equation 11)
3:	**Find** 2 parameters:
	*d* (sorting *N* subcarriers based on maximizing $\Gamma_{n}$)
	d1 (sorting *N* subcarriers based on the minimum SU $d_{c1,n}$ that accesses the $n^{\mathrm{th}}$ subcarrier
4:	$\forall SU_{j}$, find the $d_{v}$ subcarrier based on *d*
	$\forall d_{v,n}\in d$ according to the requirements
5:	**If**$\Gamma_{v,n}>\Gamma_{th}$, **and** (Equation 7)
6:	Allocate the $n^{\mathrm{th}}$ subcarrier to $SU_{j}$ based on *d*
7:	Calculate: Interference on each $n^{\mathrm{th}}$ subcarrier based on (Equation 9)
8:	Update ($\Gamma_{n}$) (Equation 11)
9:	**else if** (Equation 7) is not fulfilled
10:	**Look for** the available subcarrier based on d1
11:	**If** (Equation 7) for subcarrier based on d1 is fulfilled.
	**and** $\Gamma_{n(d1)}>\Gamma_{th}$
12:	Allocate the $n^{\mathrm{th}}$ subcarrier to the SU based on d1
13:	Calculate: Interference on each $n^{\mathrm{th}}$ subcarrier based on (Equation 9)
14:	Update ($\Gamma_{n}$) (Equation 11)
15:	**end if**
16:	**End if** $\sum_{SU,n}=d_{c}$ **or** $\Gamma_{n}\leq \Gamma_{th}$
	**or**∀SU have been allocated.

**Table 3 sensors-19-01921-t003:** Simulation parameters.

Parameters	Type/Value
Path loss models	Cost231-wi NLOS
Number of PUs (*I*)	10
Number of SUs (*J*)	20–200
Total number of subcarriers	100
Number of subcarriers of the PU ($N_{PU}$)	10
$d_{v}$	4, 8, 12, and 16
$d_{c}$	3, 6, 9, and 12
Transmission power of the PU ($P_{PU}$)	23 dBm
Transmission power of the SU ($P_{SU}$)	10 dBm
$P_{PU}$ in each subcarrier	$\frac{P_{PU_{i},n}}{N_{PU}}$
$P_{SU}$ in each subcarrier	$\frac{P_{SU_{j},n}}{d_{v}}$
Noise spectral density ($\sigma^{2}$)	−300 dBm/Hz
Bandwidth of each subcarrier ($W_{n}$)	15 kHz

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
