# Peer review of "Radio Resource Allocation with The Fairness Metric for Low Density Signature OFDM in Underlay Cognitive Radio Networks"

_sensors, 2019, doi:10.3390/s19081921_

Reviewer 1 Report

Please find below my comments/suggestions.

Carefully read the manuscript to remove typos error.

Explain ILRA-FM algorithm clearly in body text.

Add the operational example for proposed schemes and explain it using a Timing diagram. Please see the following article "Opportunistic channel selection MAC protocol for cognitive radio ad hoc sensor networks in the internet of things,Sustainable Computing: Informatics and Systems, https://doi.org/10.1016/j.suscom.2017.07.003"

Add detail simulation parameters in tabular form.

Results are not properly justified. Explain all figures/sub-figures with proper reasoning to justify the results. 

Author Response

Response to Reviewer 1 Comments

Point 1: Carefully read the manuscript to remove typos.

Response 1: We have read the manuscript iteratively and made several improvements corrections to eliminate typos. Several typos that have been corrected as follows:

-        lines 9-11, we have already corrected the sentence “Simulation results show  that ILRA-FM algorithm has the lower outage probability and higher fairness metric and also higher fairness index than interference limit based resource allocation (ILRA).” Becomes “Simulation results show that compared to ILRA algorithm, ILRA-FM algorithm has a lower outage probability and higher fairness metric value and also higher throughput fairness index.”

-        Line 19: “… in PU performance” becomes “PU’s performance”.

-        Line 54: “…to increase the fairness of allocating dv subcarrier to each user.” change to be “…to increase the fairness of allocating dv subcarriers to each user.”

-        Line 69: “where the channel of j-th PU and i-th SU at n-th subcarrier” becomes  “where the channel of jth PU and ith SU at nth subcarrier”

-        Line 99: “The rate of k-th PU n-th subcarrier…” becomes  “The rate of ith at nth subcarrier…”

-        Line 112: “j-th” becomes “jth”

-        Table 2: “Inisiation” has been corrected to be “Initiation”

-         

Point 2: Explain ILRA-FM algorithm clearly in body text.

Response 2: We have added step-by-step explanation of ILRA-FM algorithm in body text (lines 124-133) and added explanation about ILRA-FM type 2 lines 146-150 in the paper. The correction that we made on the paper is copied here as follows:

ILRA-FM algorithm (lines124-133):

The first step in ILRA–FM algorithm is setting the value of all parameters. The second step is initiation process. In this step all of PUs access their channels and CR system will calculate the SINR for each subcarrier based on (11) by assuming there is no SUs in each subcarrier (Ii,n = 0). By using two parameters (d and d1) in ILRA–FM algorithm, CR determine the number of dv subcarriers that have higher SINR and check for the constraint (7). If the constraint (7) has been fulfilled, system will allocate dv subcarriers to SU, calculate the interference in each subcarrier based on (9) and update the SINR value for each subcarrier based on (11). However, if one or several selected subcarriers (based on d parameter) do not meet the constraint (7), the system will search for available subcarriers that meets the constraint (7) based on d1, check the constraint (9) for each selected subcarrier, allocate the subcarriers to SU and update the SINR value.

Lines 145-149:

The two types of ILRA–FM algorithm have the same goal, i.e. to minimize the outage probability and increase the fairness index. Similar to ILRA–FM type 1, ILRA–FM type 2 also use two parameters, d and d1. In ILRA–FM type 2, the selected subcarrier from d1 is always compared to the selected subcarrier from d in terms of number of users accessing it. Therefore, the allocated subcarriers are not always prioritized based on the highest SINR.

Point 3: Add the operational example for proposed schemes and explain it using a Timing diagram. Please see the following article “Opportunistic channel selection MAC protocol for cognitive radio ad hoc sensor networks in the internet of things, Sustainable Computing: Informatics and Systems, https://doi.org/10.1016/j.suscom.2017.07.003”

Response 3: We sincerely thank you for suggesting the reference. We download the paper and included in our reference list [17]. In respond to that, we have added an operational example of ILRA-FM type 1 in body text (see lines 133-145) and we also develop a timing diagram of ILRA-FM algorithm in Figure 4 (see page 7). We need emphasize that the timing diagram in that reference shows the channel utilization interval information, successful data frame transmission based on data frame transmission table, while the timing diagram of our proposed ILRA-FM algorithm provides information about the subcarrier accessed by each SU based on the selected subcarrier (based on d and d1). The revised version of the paper in accordance with point 3 comment/question (lines 133-145, and the timing diagram) is copied here as follows:

Figure 4 illustrates a timing diagram for operational example of ILRA-FM algorithm when dv = 2 and dc = 3. The value "1" in the allocated subcarriers indicates that the subcarriers are allocated to SU. At t = t0, system calculate the SINR in each subcarrier. Based on highest SINR value, d parameter will choose dv subcarriers, in Figure 4, the 3rd and the 4th subcarriers are selected, the 1st and the 2nd subcarriers are selected based on parameter d1. Although there are 4 different subcarriers are selected based on d and d1, ILRA–FM algorithm always choose dv subcarriers based on d for the first step, except the selected subcarriers do not edequite to (7). Therefore, at t = t1, the 3rd and the 4th subcarriers are allocated to SU1, while based on selected subcarrier at t = t5, the 1st and the 2nd subcarriers are allocated to SU6. Although the 3rd subcarrier is selected (based on d) however it is not appropriate to constraint (7), cause, it has been accessed by three different users SU1, SU2, and SU4. Due to the value of dc parameter is equal to 3, consequently, SU8 is allowed to access one subcarrier only (2nd subcarrier), and SU9 is not allowed to access any subcarrier.

Point 4: Add detail simulation parameters in tabular form.

Response 4:  Simulation parameters have been tabulated as requested, and it is presented in Table 3 on page 7. The table is copied here as follows:

Point 5: Results are not properly justified. Explain all figures/sub-figures with proper reasoning to justify the results.

Response 5: We have rewritten description of results presented in Figures 5-12 to justify its reasoning. Justification of results that was presented on Figures/sub figures were discussed directly right after each individual figures/sub figure or at least in the same page/same section. We explain our reasoning of Figure 5 in lines 167-189; reasoning of Figure 6 in lines 190-197; reasoning of Figures 7, 8 and 9 in lines 199-210; Figures 10, 11, and 12 in lines 211-229. For better clarity and easier reading, we splitted Figure 6 of our previous version into 3 figures (figures 7, 8, and 9), and we splitted figure 7 into figure 10, 11, 12, in our new revised version of the manuscript.  Here, we copied our revision as follows:

Reasoning of Figure 5 (lines 167-189):

Figure 5 shows the outage probability, number of SUs that obtain the resource allocation less than dv subcarriers, the fairness metric and the throughput fairness index for ILRA algorithm, ILRA–FM algorithm type 1 and 2 when all algorithm setting the same value value of dv and dc. Figure 5 (a) shows the proposed algorithm (ILRA–FM-1) and (ILRA–FM-2) has lower outage probability than ILRA algorithm as proposed in [15]. The addition of sorting parameter d1 in ILRA–FM algorithm causes the algorithm to ensure each SU to acquire dv subcarriers by searching for alternative subcarriers accessible for SU when the selected subcarriers based on SINR do not fulfill the constraint in (7). Consequently, a number of SUs obtaining the resource allocation will increase and cause the decreasing outage probability (as shown in Figure 5 (a)), increase the fairness metric (Figure 5 (c)) and increase throughput fairness index ( shown in Figure 5 (d)). Figure 5 (b) states the comparison of the number of users with resource allocation less than dv subcarriers to the total number of users in the system. This figure 178 shows that ILRA FM types 1 and 2 have better performance than ILRA in terms of allocating dv subcarriers on each SU. However ILRA–FM type 2 has a lower performance than type 1 in the number of subcarriers allocated to each SU. ILRA–FM type 2 always compares the selected subcarriers based on the parameters d and d1 in terms of the number of users who have accessed the subcarrier. When the selected subcarriers of the two parameters differ in sequence, for example the selected subcarrier based on d is SC1 (accessed by 2 users), SC3 (accessed by 4 users) and the selected subcarrier based on d1 is SC2 (accessed by 3 users), SC1 (accessed by 2 user) will be allocated to jth SU. In the case of dv = 2, ILRA–FM type 2 will compare SC1 to SC2 and choose SC1 because SC1 only accessed by 1 user and update the number of users accessing SC1 to 3 users, when ILRA–FM type 2 compare the next selected subcarrier, SC3 and SC1, the algorithm still chooses SC1 and causes the number of subcarriers allocated to jth SU only SC1. However, ILRA-FM type 2 still has a better value of fairness metric than ILRA as shown in Figure 5 (c).

Reasoning of Figure 6 ( lines 190-197):

Figure 6 (a) and (b) show the average throughput both of PUs and SUs. From the figures, it can be seen that ILRA–FM algorithm has lower throughput value than ILRA algorithm. The low throughput value in the ILRA–FM algorithm is caused by the higher interference value perceived by each user, both PUs and SU as a result of the increase in number of SUs allocated in each resource. The average throughput shown in Figure 6 only takes into account the average throughput of allocated users, while the throughput fairness index in Figure 5 (d) consider the throughput of all SUs. Therefore, The proposed ILRA–FM algorithm type 1 and 2 in Figure 5 (d) have higher fairness index than ILRA algorithm.

Reasoning of Figures 7, 8 and 9 (lines 199-210):

Figures 7, 8 and 9 show the effect of different value of dc when system set the fixed value of dv. Setting fixed value on dc in LDS–OFDM system causes the number of SUs who can access each resource to be limited to dc user, therefore, the smaller value of dc will increase the outage probability and higher dc will cause more SU users allowed to access the subcarrier. Therefore, the outage probability will decrease, fairness metric and throughput fairness index will increase. Figure 7 proves that higher value of dc has minimum OP compare to another value of dc (4 and 12). Figure 8 (a) and (b) have shown that higher value of dc cause higher FM and FI as well. The smaller value of dc also cause both of PU’s and SU’s transmission rate higher due to low interference felt by each user. Figure 9 (a) and (b) show the average throughput both of PUs and SUs. From these figures, it can be stated that higher value of dc will cause lower throughput both of PUs and SUs. As the higher number of SUs accessed each subcarrier, the interference that felt by each user, both PUs and SUs, will be higher. Based on (4) and (14), higher the interference will reduce the throughput in each subcarrier.

Reasoning of Figures 10, 11, and 12 (lines 211-229):

Figure 10 shows the effect of dv parameter in the outage probability. Figure 11 shows the effect of  dv parameter in fairness metric and throughput fairness index. While 12 shows the effect of dv parameter in the average throughput PU and SU. The higher value of dv causes the higher outage probability, lower fairness value on FM and FI and also lower average throughput both PUs and SUs. The dv parameter states the number of subcarriers accessed by each SU and determine SU’s transmission power in each subcarrier. When the dv parameter is set high, then each SU will spread their symbols to more subcarriers and cause the number of users accessing the subcarrier to reach the limit faster and then the outage probability will increase and both of fairness value will decrease. Due to the higher value of dv, SU’s transmission power on each subcarrier will be lower. Although the higher value of dv cause the number of users accessing the subcarrier much closer to dc, due to lower transmission power in each subcarrier, SU’s transmission rate will be lower. The effect of dv parameters on average throughput is shown in Figure 12. Figure 12 (a) shows average throughput of PU. When the number of active SU on the system is in the range of 20 to 100 users, the higher dv value causes the lower throughput of PU. Conversely, when the number of users is greater than 100 users the lower dv value causes lower throughput of PU. The decreasing the average throughput of PU is caused by the increased interference felt by PU. When dv is set small, the transmission power of SU in each allocated subcarrier will be greater. Figure 11 (a) shows fairness metric values are almost close to 1 when the number of users is 100 and dv = 4, meaning that each SU will attain the same number of subcarriers. Because of system set dc = 9, each subcarrier is accessed by a dc user (9 users),  means that there are 9 SUs with higher transmission power access the same subcarrier.

Reviewer 2 Report

Abstract:

"Simulation results show that ILRA-FM algorithm has the lower outage probability and higher fairness metric and also higher fairness index than interference limit based resource
allocation (ILRA)."

Lower compared to what?

Please explain how your work differentiates from [11] better and in more detail.

Please explain how this work differentiates from your work of

"Dual List Interference Cancellation in Underlay
Cognitive Radio"

Please explain the algorithm at table 1 more in detail.

Author Response

Response to Reviewer 2 Comments

Point 1: Abstract:” Simulation results show that ILRA-FM algorithm has lower outage probability and higher fairness metric and also higher fairness index than interference limit based resource allocation (ILRA).” Low compared to what?

Response 2: We have rewritten this sentence in lines 9-11 to:

Simulation results show that compared to ILRA algorithm, ILRA-FM algorithm has a lower outage probability and higher fairness metric value and also higher throughput fairness index.

Point 2: Please explain how your work differentiates from [11] better and in more detail.

Response 2: Thank you for the suggestion. In our revise manuscript we have added some explanation about ILRA lines 101-114 in body text. We also explained our proposed algorithm in more detail by adding some explanation in body text lines 124-133, lines 146-150, taking into account your suggestion and suggestion from Reviewer 1. We added an operational example for ILRA-FM in lines 133-145 and add a timing diagram for ILRA-FM algorithm in Figure 4 in page 7.  Our simulation results in Figure 5(c) has shown that both of ILRA-FM type 1 and 2 has higher value of fairness metric rather than fairness metric of ILRA. It means that all SU in ILRA-FM has allocated subcarrier close to dv value. The correction that we made on the paper is copied here as follows:

ILRA algorithm (lines 101-114):

2.3. Interference Limit Based Resource Allocation (ILRA)

Reference [15] has already allocated LDS–OFDM users based on interference limit, we named it as interference limit-based resource allocation algorithm (ILRA) algorithm. ILRA algorithm allocate a number of resources for each SU based on lower interference value or higher SINR in each subcarrier. ILRA algorithm is showed in Table 1.

In the first step, we must set the value of dv, dc, SINR threshold, a total number of subcarriers, N, a number of subcarrier accessed by each PU, NPU, total number of PUs, I, total number of SUs, J, transmission power both of PU and SU, PPU and PSU. After all PUs have accessed their subcarriers, the CR system will calculate interference or SINR in each subcarrier and sort the subcarriers based on the highest SINR. Select dv subcarriers with highest SINR to be allocated to the jth SU provided that the subcarriers meet requirement (7). If there are dx subcarriers in dv selected subcarriers that has been accessed by dc users, the jth SU will only obtain an allocation of dv- dx subcarriers. Hence, ILRA algorithm cause number of the resource allocation for jth SU less than dv subcarriers.

ILRA-FM algorithm (lines124-133):

The first step in ILRA–FM algorithm is setting the value of all parameters. The second step is initiation process. In this step all of PUs access their channels and CR system will calculate the SINR for each subcarrier based on (11) by assuming there is no SUs in each subcarrier (Ii,n = 0). By using two parameters (d and d1) in ILRA–FM algorithm, CR determine the number of dv subcarriers that have higher SINR and check for the constraint (7). If the constraint (7) has been fulfilled, system will allocate dv subcarriers to SU, calculate the interference in each subcarrier based on (9) and update the SINR value for each subcarrier based on (11). However, if one or several selected subcarriers (based on d parameter) do not meet the constraint (7), the system will search for available subcarriers that meets the constraint (7) based on d1, check the constraint (9) for each selected subcarrier, allocate the subcarriers to SU and update the SINR value.

Lines 145-149:

The two types of ILRA–FM algorithm have the same goal, i.e. to minimize the outage probability and increase the fairness index. Similar to ILRA–FM type 1, ILRA–FM type 2 also use two parameters, d and d1. In ILRA–FM type 2, the selected subcarrier from d1 is always compared to the selected subcarrier from d in terms of number of users accessing it. Therefore, the allocated subcarriers are not always prioritized based on the highest SINR.

Operational example of ILRA-FM algorithm (lines 133-145):

Figure 4 illustrates a timing diagram for operational example of ILRA-FM algorithm when dv = 2 and dc = 3. The value "1" in the allocated subcarriers indicates that the subcarriers are allocated to SU. At t = t0, system calculate the SINR in each subcarrier. Based on highest SINR value, d parameter will choose dv subcarriers, in Figure 4, the 3rd and the 4th subcarriers are selected, the 1st and the 2nd subcarriers are selected based on parameter d1. Although there are 4 different subcarriers are selected based on d and d1, ILRA–FM algorithm always choose dv subcarriers based on d for the first step, except the selected subcarriers do not edequite to (7). Therefore, at t = t1, the 3rd and the 4th subcarriers are allocated to SU1, while based on selected subcarrier at t = t5, the 1st and the 2nd subcarriers are allocated to SU6. Although the 3rd subcarrier is selected (based on d) however it is not appropriate to constraint (7), cause, it has been accessed by three different users SU1, SU2, and SU4. Due to the value of dc parameter is equal to 3, consequently, SU8 is allowed to access one subcarrier only (2nd subcarrier), and SU9 is not allowed to access any subcarrier.

Point 3: Please explain how this work differentiates from your work of “Dual List Interference Cancellation in Underlay Cognitive Radio”.

Response 3:  Our previous work “Dual List Interference Cancellation and Underlay Cognitive Radio”, discuss about spectrum access in underlay scheme and interference cancellation technique. This technique uses dual list of signals: signal of interest and interference signal, combine and modified successive interference cancellation (SIC) and Parallel interference cancellation (PIC). We use multi carrier code division multiple access (MC-CDMA) as a multiple access technique. Both PUs and SUs interfere each other in all subcarrier, and the system is in under load condition. Our proposed work in “Radio Resource Allocation with Fairness Metric for Low Density Signature OFDM in Underlay Cognitive Radio Networks” still discuss in underlay scheme of Cognitive radio. However, we use another approach to interference problem. In this paper we do not discuss how to mitigate /cancell the interference but use one type of code domain non-orthogonal multiple access (NOMA), i.e. low density signature orthogonal frequency division multiple access (LDS-OFDM) to manage the interference by allocate SU in a number of subcarriers and limit a number of users accessing each resource. The important point in this paper is fairness in allocating dv subcarriers for each SU. In this paper we assume that all SUs have the same priority and and will be allocated in the same number of resources based on interference limit and fairness metric. In revised version we added some explanation about our previous work in lines 23-32 in body text of paper. The explanation about our previous work in the revised version is copied here as follows:

Lines23-32:

There are many studies related to spectrum access and interference mitigation in CR systems. Due to the flexibility in allocating resource among SUs, many researchers have used multi–carrier transmission in CR networks. References in [2–5] used multi carrier code division multiple access (MC–CDMA) as their spectrum access model and designed interference cancellation (IC) techniques to combat the interference. Reference [4] assumed that each PU and SU interfered each other. To combat the interference both in PUs and SUs [4] used dual list interference cancellation (DLIC), as a combination of successive interference cancellation (SIC) and parallel interference cancellation (PIC). Unlike the previous work in [4], in this paper we use low density signature orthogonal frequency division multiple access (LDS–OFDM) as a multiple access system and manage the interference felt by each SU to increase the spectrum utilization.

Point 4:  Please explain the algorithm at table 1 in more detail.

Response 4: We added more detail explanation about our proposed algorithm in body text lines 124-133, lines 146-150, and added operational example for ILRA-FM type 1 in lines 133-145.

ILRA-FM algorithm (lines124-133):

The first step in ILRA–FM algorithm is setting the value of all parameters. The second step is initiation process. In this step all of PUs access their channels and CR system would calculate the SINR for each subcarrier based on (11) by assuming there is no SUs in each subcarrier (Ii,n = 0). By using two parameters (d and d1) in ILRA–FM algorithm, CR determine the number of dv subcarriers that have higher SINR and check for requirement (7). If the requirement in (7) has been fulfilled, system will allocate dv subcarriers to SU, calculate the interference in each subcarrier based on (9) and update the SINR value for each subcarrier based on (11). However, if one or several selected subcarriers (based on parameter d) do not meet the requirement (7), the system will search for available subcarriers that meets requirement (7) based on d1, check the requirement (9) for each selected subcarrier, allocate the subcarriers to SU and update the SINR value.

Lines 146-150:

The two types of ILRA–FM algorithm have the same goal, i.e. to minimize the outage probability and increase the fairness index. As well as ILRA–FM type 1, ILRA–FM type 2 also use two sorting parameters d and d1. In ILRA–FM type 2, the selected subcarrier from d1 is always compared to the selected subcarrier from d in terms of number of users accessing it. Therefore, the allocated subcarriers are not always prioritized based on the highest SINR.

Operational example of ILRA-FM algorithm (lines 133-145):

Figure 4 illustrates a timing diagram for operational example of ILRA-FM algorithm when dv = 2 and dc = 3. The value "1" in the allocated subcarriers indicates that the subcarriers are allocated to SU. At t = t0, system calculate the SINR in each subcarrier. Based on highest SINR value, d parameter will choose dv subcarriers, in Figure 4, the 3rd and the 4th subcarriers are selected, the 1st and the 2nd subcarriers are selected based on parameter d1. Although there are 4 different subcarriers are selected based on d and d1, ILRA–FM algorithm always choose dv subcarriers based on d for the first step, except the selected subcarriers do not edequite to (7). Therefore, at t = t1, the 3rd and the 4th subcarriers are allocated to SU1, while based on selected subcarrier at t = t5, the 1st and the 2nd subcarriers are allocated to SU6. Although the 3rd subcarrier is selected (based on d) however it is not appropriate to constraint (7), cause, it has been accessed by three different users SU1, SU2, and SU4. Due to the value of dc parameter is equal to 3, consequently, SU8 is allowed to access one subcarrier only (2nd subcarrier), and SU9 is not allowed to access any subcarrier.

Round  2

Reviewer 1 Report

I am happy with the revised version.